# Flavor Wheel Construction and Sensory Profile Description of Human Milk

**DOI:** 10.3390/nu14245387

**Published:** 2022-12-19

**Authors:** Mingguang Yu, Chengdong Zheng, Qinggang Xie, Yuan Tang, Ying Wang, Baosong Wang, Huanlu Song, Yalin Zhou, Yajun Xu, Rongqiang Yang

**Affiliations:** 1Laboratory of Molecular Sensory Science, School of Food and Health, Beijing Technology and Business University (BTBU), Beijing 100048, China; 2Heilongjiang Feihe Dairy Co., Ltd., C-16, 10A Jiuxianqiao Rd., Chaoyang, Beijing 100015, China; 3PKUHSC-China Feihe Joint Research Institute of Nutrition and Healthy Lifespan Development, Xueyuan Road 38, Haidian, Beijing 100083, China; 4Department of Nutrition and Food Hygiene, School of Public Health, Peking University, Xueyuan Road 38, Haidian, Beijing 100083, China; 5Beijing Key Laboratory of Toxicological Research and Risk Assessment for Food Safety, Peking University, Xueyuan Road 38, Haidian, Beijing 100083, China; 6Department of Clinical Nutrition, Anhui Provincial Children’s Hospital, Wangjiang East Road 39, Hefei 230000, China

**Keywords:** human milk, flavor wheel, sensory lexicon, descriptive analysis, sensory characteristics, sensory profile descriptors

## Abstract

To explore the flavor characteristics of human milk, we constructed a three-tiered human milk flavor wheel based on 53 sensory descriptors belonging to different sensory categories. Fifteen sensory descriptors were selected using M-value and multivariate statistical methods, and the corresponding references were set up to realize qualitative and quantitative sensory evaluation of the human milk samples. To ensure the accuracy and reliability of the sensory evaluation, the performance of the sensory panelists was also tested. The sensory profile analysis indicated that the established sensory descriptors could properly reflect the general sensory properties of the human milk and could also be used to distinguish different samples. Further investigation exposed that the fat content might be an important factor that influence the sensory properties of human milk. To the best of our knowledge, this is the first report on the flavor wheel of human milk.

## 1. Introduction

Human milk is called the “Life liquid” with complex components, inter-individual differences, and various nutrients [1]. It is considered the best natural food for infants, because it contains specific nutrients and bioactive factors that provide physiological, cognitive, and emotional benefits for infants [2]. Compared with the infant formula feeding mode, breastfeeding is widely considered to be the best feeding mode for infants [3]. It not only affects the growth and development of infants but also has an important impact on the health of both infants and mothers [4]. Therefore, researchers have made tremendous efforts to study the functions of various components in human milk; however, less attention has been paid to the flavor of human milk. Existing studies have confirmed that the flavor of human milk plays a critical role in breastfeeding. For example, it can guide infants to recognize human milk [5], promote their sucking behavior and milk intake [6], and pacify infant emotions [7]. Even the flavor of human milk can influence an infant’s future eating preferences [8]. Therefore, it is essential to comprehensively analyze and deeply explore the flavor and sensory characteristics of human milk.

Sensory evaluation is a statistical analysis method based on human sensory perception, which is a key technology for evaluating food flavor quality and sensory characteristics [9]. Sensory evaluation can be categorized as hedonic or analytical type according to sensory participants [10]. Among them, descriptive analysis (analytical type) has been widely used to describe the sensory characteristics of food and to obtain an in-depth understanding of the flavor differences between food samples [11]. Sensory evaluation involves the qualitative and quantitative description of sensory attributes of the product by a sensory panel [12]. Using quantitative descriptive analysis, it is possible to describe the product with the minimum number of terms (sensory profile descriptors) to provide the maximum amount of information [13]. Moreover, the sensory profile descriptors with reference standards can objectively describe the sensory characteristics of products. In short, the generation of sensory profile descriptors for food or products is accompanied by the screening of appropriate sensory panel and the development of standardized protocols for evaluators to follow [14]. At the same time, the generated sensory profile descriptors need to be defined and the corresponding reference standards need to be attached [9]. Flavor is an important factor influencing infants’ food choices, and human milk, as a special food for infants, is more conducive to finding suitable substitutes by exploring its flavor properties. It should be especially noted that infants may have a more sensitive sense of smell than adults, but sensory evaluation through infants is somewhat unworkable. Therefore, sensory evaluation at the adult level should be conducted to obtain consistency in flavor of alternatives with human milk, and then further refine the flavor of alternatives through sensory selectivity of infants.

As a practical visual tool for describing flavor characteristics of food products, the flavor wheel can represent the flavor characteristics of tested samples by collecting, classifying, summarizing, and sorting specific sensory attribute descriptors [11]. It is an important means of standardization of quantitative descriptive analysis in the food field and a reliable basis for sensory analysis and communication between producers and consumers [15]. Overall, the flavor wheel was developed based on the sensory evaluation results by a professional sensory panel [16]. The accurate and effective sensory descriptors are screened by multivariate statistics. Then, the descriptors are divided into 2 to 3 levels to summarize. The first-level terms belong to the macro category and are usually divided according to the recognition mode (such as aroma, flavor, mouthfeel, and texture). The second-level terms are refined and classified according to first-level terms. The third-level terms refer to concrete object descriptors. Finally, the above three levels of descriptors are represented by a circular wheel image [17]. Whether it is typical style characteristics or off-flavor defects, the flavor characteristics of samples can be visually displayed by the flavor wheel. This enables manufacturers to better control product quality and lays the foundation for further flavor discovery and research, which supports product improvement and new product development. At present, the flavor wheel has been applied in many food fields, such as tea [18], coffee [19], wine [20], chocolate [21], and other fields, in which have built their unique flavor wheels. However, there are few studies on dairy products, especially on human milk.

Based on ISO 11035, this study explored the sensory properties of human milk and visualized them into flavor wheels by grouping these sensory properties. At the same time, the sensory profile descriptors of human milk were established to provide a scientific evaluation method to accurately describe and distinguish the different sensory characteristics of human milk samples. In addition, by exploring the characteristics and differences of sensory attributes of different human milk samples, this study also aimed to uncover the potential factors that might lead to the differences in the flavor of human milk.

## 2. Materials and Methods

### 2.1. Donors

A total of 18 mothers with age range 25–35 (mean age 28) were recruited as donors. All of them were in a lactation period of 1 to 6 months postpartum and delivered healthy full-term offspring. These donors came from Beijing, Jiangsu and Anhui in China, with six of them from each of the above regions. All of them had no habit of smoking or alcohol drinking, were healthy, had no history of medication during breastfeeding, and had regular diet and lived in a stable environment. The production of human milk was normal and exceeded the feeding needs of their infants, and there were no breast infections. Human milk samples were collected from 9 to 11 a.m., which was approximately 2 h after the donor’s last breastfeeding, using their own breast pump on one side, from which the milk was completely expressed to obtain a total sample volume of 100–150 mL, collected as mixtures, including foremilk and hindmilk, to simulate the most common maternal breastfeeding behaviors [22]. This research was affiliated with an observational study ( https://clinicaltrials.gov/ct2/show/NCT05133466 (accessed on 24 November 2021), identifier: NCT05133466, 17 September 2021) sponsored by Heilongjiang Feihe Dairy Co., Ltd. and was ethically approved by The Ethics Committee of Peking University (Approval Number IRB00001052-21091). Informed consent forms were carefully read and signed by all human milk donors after the purpose and nature of the study were fully explained.

### 2.2. Samples

To capture as many potential sensory descriptors as possible, three regions were selected for human milk collection, and a total of 18 human milk samples were obtained (M1~M18). Samples were transported immediately to the laboratory at −78 °C with dry ice after collection. Before the sensory test, all samples were stored in the freezer at −80 °C in independently sealed packaging bags. The human milk compositions were rapidly determined using MirisHMA Human milk Analyzer (Uppsala, Sweden). The sample information and composition are shown in Table 1. To help to understand the effect of regions, lactation periods and total fat contents on the flavor of human milk, samples obtained were accordingly categorized: high lactation period (LP-H, 5–6 months), medium lactation period (LP-M, 3–4 months), and low lactation period (LP-L, 1–2 months); high total fat content (TFC-H, >4 g/100 mL), medium total fat content (TFC-M, 2.5~4.0 g/100 mL), and low total fat content (TFC-L, <2.5 g/100 mL); Beijing (BJ), Jiangsu (JS), and Anhui (AH).

### 2.3. Sensory Evaluation Panel Training

According to the methods of selection of sensory evaluators (GB/T 16291.1-2012), 16 candidate sensory evaluators were selected from the molecular sensory laboratory of Beijing Technology and Business University with ages ranging from 24 to 30 years old (mean age 27). Candidates gave informed consent via the statement “I am aware that my responses are confidential, and I agree to participate in this survey”. They were able to withdraw from the survey at any time without giving a reason. The products tested were safe for consumption. The candidates were healthy and with no habits of smoking or alcohol drinking. These candidate sensory evaluators have participated in sensory evaluation tests of human milk [23], yogurt [24], and infant formula [25]. All of them had at least two years of experience in sensory evaluation of dairy products. Then, the sensory function (smell and taste), sensory sensitivity, and descriptive ability of the candidate sensory evaluators were further tested in an odor-free sensory analysis laboratory. A total of 12 sensory evaluators (five men and seven women) were selected to form a sensory panel. Subsequently, the panelists received sensory training according to the consensus method described by Lawless and Heymann [11,26]. The training was held five times a week, two hours each time and lasted for one month.

### 2.4. Sensory Lexicon Generation and Screening

All samples of human milk were coded with a 3-digit random code. Each of the human milk samples (15 mL) for sensory testing was stored in a 50 mL glass bottle and heated to 37 °C in a water bath. Firstly, six human milk samples were randomly selected for evaluation by each panelist. They were asked to give descriptions of human milk in terms of aroma (orthonasal evaluation), flavor (retronasal evaluation, including basic taste), and mouthfeel perception. Briefly, they were first asked to smell the human milk samples and write down as many aroma descriptors as possible to describe smell perception. The panelists were then asked to perform the same tests on the taste and mouthfeel perception of human milk. Secondly, to generate objective terms and avoid subjective terms or opinions, the above obtained sensory lexicons were preliminarily screened according to the international organization standardization of sensory descriptors selection method (ISO 11035) [27]. The hedonistic terms (such as fine, good, pleasant, etc.), quantitative terms (such as too much, strong, etc.), terms describing the product with its own terms, and irrelevant terms were deleted; the terms with similar meanings were merged. Thereby, the sensory descriptors for human milk flavor wheel were obtained (Table 2). Finally, a 6-point scale (0, none; 1, weak; 2, slightly weak; 3, average; 4, slightly strong; 5, strong) was used to evaluate the strength of the sensory properties of 18 human milk samples according to the obtained sensory lexicons. The contribution of sensory lexicons was ranked by calculating the geometric mean *M* [27]:$$M=\sqrt{F*I}$$
where, *F* represents the ratio of the times of a specific descriptor actually mentioned to its total times that possibly mentioned; *I* is the ratio of the sum of actual intensities for a descriptor given by the evaluation panel over its maximum possible intensity. The larger the value of *M*, the greater contribution of the sensory lexicons to the sensory characteristics of the sample. Each human milk sample was analyzed 3 times.

### 2.5. Construction of the Sensory Wheel

The flavor wheel was depicted from the sensory lexicons of the human milk in Table 2. The three major sensory modalities, namely aroma, flavor, and mouthfeel, were the first-level terms. It forms the innermost tier of the flavor wheel. Descriptors for 53 specific attributes were level 3 terms, located in the outermost tier of the flavor wheel. The sub-descriptors used to group specific type attributes were level 2 terms, located in the second tier, and there were 18 of them.

### 2.6. Establishment and Application of Quantitative Descriptors for Breast Milk

According to ISO 11035, the number of descriptors in the flavor wheel of human milk was further reduced: (1) Based on the geometric mean M of the descriptors in the flavor wheel, the descriptors with lower M value were firstly deleted. (2) Principal component analysis (PCA) and hierarchical cluster analysis (HCA) were carried out for reserved descriptors with large M values, and discussion was conducted by the sensory panel at the same time [27]. Descriptors with high correlation were merged and descriptors with large variance (significant difference) were retained. (3) Furthermore, definitions, the corresponding reference samples, and their intensity of the selected sensory descriptors were offered to the panel to help establish a quantitative description vocabulary of human milk (Table 3). All kinds of reference samples were stored in a 50 mL glass vial for the evaluation and training of sensory panel members.

In addition, based on the above quantitative description vocabulary of human milk, the sensory intensity of human milk was quantitatively scored on a scale of 0 to 5 with 0.5 increments, where “0” means that the flavor was not felted, and “5” means that the flavor was strong [27].

### 2.7. Sensory Panelists Performance Evaluation

Based on the established quantitative descriptive vocabulary of human milk, sensory panelists were asked to perform sensory evaluation of the corresponding attributes of three human milk samples that were randomly selected from each of the three regions. PanelCheck 1.4.0 online platform was used to evaluate the performance of sensory panelists [28].

### 2.8. Statistical Analysis

One-way analysis of variance (ANOVA) was used to analyze quantitative data from sensory panel and identify significant differences between human milk samples from different regions. The performance of sensory panels was monitored using PanelCheck (Version 1.4.0, http://www.panelcheck.com). PCA and HCA were used for secondary screening of sensory descriptors. Other data processing was completed by Microsoft Office Excel 2016 software (Microsoft^®^ Excel^®^ 2016MSO (version 2211 Build 16.0.15831.20098) 64-bit).

## 3. Results and Discussion

### 3.1. Construction of Breast Milk Flavor Wheel

The human milk samples were provided to the sensory panel, and the sensory descriptions of aroma (orthonasal evaluation), flavor (retronasal evaluation, including basic taste), and mouthfeel were analyzed, and sensory descriptors were proposed. A total of 84 descriptors were generated. These descriptors were preliminarily collated by the sensory panelists, removing pleasurable terms (such as unpleasant, bad, etc.) and quantitative terms (such as light cooking smell, light milk smell, strong fishy smell, etc.). At the same time, the terms with opposite meanings (such as smoothness/smooth and coarse/grainy, etc.) were re-evaluated, discussed, and screened. Repeatedly described terms or the ones with similar meanings were merged and the descriptors with a frequency mentioned by the panelists of less than 5% were deleted [18]. Finally, 30 aroma descriptors, 14 flavor descriptors and nine mouthfeel descriptors were selected from the list of descriptors generated by the sensory evaluation team, altogether 53 human milk sensory descriptors (Table 2).

Referring to the establishment of the lexicon of milk powder [29], soy milk [30], pure milk [31], and other samples, the descriptors of aroma, flavor, and mouthfeel were classified by sensory panel discussion, which constituted the second-level terms of flavor wheel. According to classification, the second-level terms of aroma can be divided into 10 categories, including dairy, earthy, fatty-acid [32], sweet, vegetable, fishy, fruity, floral, bakery [21], and other categories. The second-level terms of flavor included dairy, basic taste, sweet and earthy. In addition, mouthfeel was composed of five second-level terms: silky, fattiness, astringency, mouthfulness, and rough [33]. Finally, the flavor wheel of human milk was plotted according to the above selected descriptors and methodology (Figure 1) and was composed of three main sensory modalities, namely aroma, flavor, and mouthfeel, with a total of three layers that contained 3, 18, and 53 terms from the inside out.

### 3.2. Screening of Breast Milk Sensory Profile Descriptors

The construction of the milk flavor wheel provides a basis for the quantitative description of human milk flavor. Some of the main terms in the human milk flavor wheel can be used to describe the sensory profile of human milk flavor, which is helpful in describing the basic sensory characteristics of human milk. Since the *M* value of the descriptors can reflect their importance to the sample, the aroma, flavor and mouthfeel descriptors with *M* value higher than the mean value (i.e., 0.2623, 0.2705 and 0.4513) were retained. Among them, there were 11 aroma descriptors, five flavor descriptors, and six mouthfeel descriptors (Table 2, bolded descriptors). Furthermore, the variance of these descriptors was calculated. The larger the variance was, the more significant the difference in flavor described by the descriptors of different human milk samples; therefore, different human milk samples could be effectively distinguished. Among the aroma descriptors in Table 2 (bolded descriptors), the *M* value and variance of sweaty and rancid were relatively low, indicating that the importance and difference of these two descriptors were relatively insignificant, so they were excluded first. In general, the total amount of the descriptors should not exceed 15 in order to obtain an operable sensory profile map (ISO 11035) [27]. Therefore, the above descriptors need to be further analyzed to identify sensory profile descriptors that can describe and distinguish the basic sensory characteristics of different human milk samples.

PCA and HCA were performed for the nine aroma descriptors screened according to M value and variance (Figure 2A). As can be seen from Figure 2A, 81.10% of total variability contained in the dataset can be explained by first two principal components. This indicated that these nine descriptors covered most information of the aroma variables of human milk samples and could largely characterize the aroma characteristics of human milk. The results of HCA (Figure 2B) showed that the aroma descriptors of cooked-milk-like and egg-white-like could be grouped together, and they were finally termed as cooked after discussion by the sensory panel. Similarly, milky-fishy and beany-fishy could be merged as the descriptor of fishy (aroma). Although the aroma of flour, metallic/iron, and dairy-sweet were also assembled, the sensory panel discussion concluded that they represented different aroma attributes and therefore could not be merged. Therefore, dairy fat, dairy-sweet, fishy (aroma), metallic/iron, grass/green, flour, and cooked aroma were selected as the descriptors for quantitative evaluation of the aroma profile of human milk.

Similarly, in terms of flavor and mouthfeel of human milk (Figure 2C,E), the accumulative variance contribution rate of the first two principal components reached 98.42% and 85.80%, respectively, covering most variable information of flavor and mouthfeel of the human milk samples. These results indicated that the descriptors screened according to *M* value and variance could represent the flavor and mouthfeel characteristics of the human milk samples. The cluster analysis (Figure 2F) showed that the perception of thinness, smoothness, creaminess, and watery can be grouped into one category (silky). It was worth noting that although the flavor of fishy (flavor), umami and boiled milk were also classified into one category, they represented different flavor attributes and should not be merged (Figure 2D). Thus, the flavor of umami, creamy, fishy (flavor), boiled milk, and sweet were selected as the descriptors for quantitative analysis of the flavor profile of human milk, while silky, fattiness, and mouthfulness were selected as the ones for quantitative analysis of mouthfeel profile of human milk. In addition, the references and their preparation methods were set up for the sensory descriptors selected above for quantitative analysis of the sensory profile of human milk, and the corresponding strength of the sensory attributes were also established (Table 3).

### 3.3. Evaluation of the Sensory Panel Performance

The performance evaluation of the sensory panel was a necessary step for sample difference analysis [34]. The consistency, sample discrimination ability, and repeatability of the sensory panel were three important elements to ensure the accuracy and reliability of the sample test data [35]. Based on the sensory evaluation results of human milk sensory profile descriptors on a 6-point scale, the performance of the sensory panel was comprehensively evaluated (Figure 3).

#### 3.3.1. Consistency

Tucker-1 correlation loadings plots (Figure 3A) were used to identify descriptors with potential performance problems. As can be seen from Figure 3A, the overall performance of the 12 panelists was good for most of the descriptors, and there were significant differences in the level of 0.001 for all 15 sensory descriptors (red outer box line). Although the number of variables in this experiment was large (180, 12 panelists × 15 descriptors), the variation explained by principal components PC1 and PC2 were 52.4% and 34.1%, respectively, and the total amount of variance explained was 86.5%. This indicated that most variation in the data could be explained only by PC1 and PC2. Furthermore, the higher the concentration of points in the plots of the same descriptor, the higher the group’s consistency on the word [35]. As can be seen from Figure 3A, except for the two descriptors of dairy fat and creamy, which were relatively dispersed, the other descriptors were much concentrated, so the consistency of these descriptors among the 12 panelists was much better [36]. For the descriptors of dairy fat and creamy, the plots showed that the panelists were relatively scattered along the outer ellipse, indicating that there was some disagreement between the sensory panelists on these two descriptors.

Manhattan plots can be used to compare the systematic variation of specific descriptors across all panelists after screening through the Tucker-1 plots. The Manhattan plot (Figure 3B) confirms the phenomenon exhibited in the Tucker-1 plots above. As can be seen from Figure 3B(a,b), more than two PCs were needed for the descriptors of dairy-fat and creamy to achieve a high level of explained variance. For the other descriptors, most panelists have reached a high percentage of explained variance only by one or two PCs, except for the panelist P1 and P5 in the grassy/green and sweet descriptors, respectively, in which these two panelists were distant from the other panelists (Figure 3A). Under the premise of the same principal component, the explained variance of P5 and P1 was significantly lower than that of other panelists. For example, for the descriptor of grassy/green, the cumulative variance explained by PC1 and PC2 was greater than 80% for the other panelists as shown in Figure 3B(c). In comparison, it was only 30% for panelist P1 and it reached 70% and 90% when PC3 and PC4 were included, respectively. A similar phenomenon was observed for panelist P5 in the sweet descriptor.

#### 3.3.2. Discrimination Ability and Repeatability

He *F* and mean square error (MSE) values can be used to evaluate the discrimination ability and repeatability of sensory panel. By drawing *F* and MSE plots (Figure 3C), these two abilities of sensory panelists can be visualized. The *F* value was the ratio of inter-group differences to intra-group differences, and the larger the value is, the better the panel’s ability to distinguish related descriptors. The MSE values represent the variance within the group, and the smaller the value is, the better the repeatability. It was worth noting that the small MSE values may also be due to the panel’s failure to distinguish the samples. Therefore, the repeatability should be discussed on the basis of the panel’s ability to distinguish samples (*F* values). As shown in Figure 3C(a), the discriminating ability (*F* values) of panelists for most descriptors was close to or above the significance level of 5%, and some of them were above the significance level of 1%, such as metallic/iron, sweet, flour, fishy, and silky. This indicated that the panelists had a good ability to distinguish most descriptors. Compared with the results in the previous report [35], the *F* values in this study were relatively low, especially for aroma descriptors. This may be ascribed to the odor of human milk itself, which is low in intensity [37] and led to small differences in the sensory panel’s quantitative description scores for different aroma descriptors. In addition, almost all the MSE values for the sensory descriptors from each panelist were lower than three or even lower than one except for that of fishy (aroma) and metallic/iron (Figure 3C(b)). This showed that the panelists in this study exhibited a good repeatability in the sensory evaluation. Combined with the above results, all 12 panelists exhibited high *F* values and low MSE values for most descriptors, indicating that they had a good ability to evaluate the sensory attributes of the samples [35]. For example, panelist P2 had a relatively low *F* value but high MSE value compared to the majority of the panelists, which indicated that panelist P2 had a relatively low ability to distinguish different human milk samples and repeat the sensory evaluation results compared with the majority of the panelists. Despite that, panelist P2 still showed acceptable performance in sensory evaluation since the *F* values for most descriptors were close to or above the 5% significance level, and some even exceeded the 1% significance level.

### 3.4. Application of Sensory Profile Descriptors

The human milk samples from different regions (BJ1, BJ2, BJ3, JS1, JS2, JS3, AH1, AH2, and AH3) were quantitatively scored according to the selected sensory profile descriptors (Table 3), and the obtained sensory profile was shown in Figure 4. As can be seen from Figure 4, the tested human milk samples possessed all the descriptors in Table 3 simultaneously. This indicated that the established sensory profile descriptors of human milk could comprehensively reflect the sensory characteristics of the human milk samples. The overall scores for the aroma profile descriptors of human milk (Figure 4, displayed in orange) ranged from 0.25 to 2.83 with a mean score of 1.31. The scores for all aroma profile descriptors in human milk were less than two except for that of dairy-fat (1.38–2.83, 1.85) and dairy-sweet (1.21–2.5, 1.73). In comparison, the scores for the flavor and mouthfeel profile descriptors (Figure 4, displayed in blue and green, respectively) ranged from 1.42–4.17 and 1.17–3.33, and had mean values of 2.67 and 2.36, respectively. This indicated that most of the aroma profile descriptors in human milk had low scores compared to that of flavor and mouthfeel profile descriptors.

In all the aroma attributes, dairy fat was proved to be the most prominent one in the human milk samples, with the highest average score of 1.85, which was consistent with the results of the previous study by Zhang et al. [23]. The dairy-fat aroma could be generated by several classes of compounds, such as aliphaticaldehyde (hexanal, octanal, nonanal, etc.), polyunsaturated aldehydes ((*E*,*E*)-2,4-heptadienal, (*E*,*E*)-2,4-octadienal, (*E*)-2-decenal, etc.), ketones, and other compounds that responsible for fatty odor in human milk [38]. On the other hand, the scores of sweet in all human milk samples were higher than 3 (sweet intensity equivalent to 2.12 g sucrose /100 mL water), which was not only the most prominent attribute in the flavor part but also the strongest one in all the sensory attributes of the human milk samples. This result was consistent with the finding in former research that sweet was the dominant flavor in human milk [33]. This may be related to the fact that human milk was the richest in carbohydrates (Table 1). Because lactose is the most dominant component of human milk carbohydrates with sweet taste.

The flavor descriptors of creamy and fishy (flavor) also had high scores in all the human milk samples. However, unlike creamy, the strength of the fishy (flavor) varied widely among different human milk samples, which may be due to the wide variation in fatty acid content in different human milk samples. Similarly, the scores of most descriptors in mouthfeel were also higher than that in the aroma. Among them, the scores of silky (2.42–3.33) was the highest in all the samples, followed by mouthfulness (1.25–3.00) and fattiness (1.17–2.67), which was consistent with the results of Mastorakou et al. [33]. Therefore, the most prominent sensory attributes of human milk samples were dairy fat and dairy-sweet in aroma, sweet and creamy in flavor, and silky in mouthfeel. In other words, human milk could be characterized as a liquid with a notable dairy fat and dairy-sweet smell, sweet and creamy taste, and silky mouthfeel. These sensory characteristics were very close to our general impressions of human milk; however, it is worth noting that these may differ from those of actual fresh human milk, especially in the fishy (flavor) [22,23]. The human milk tested by the sensory panelists was frozen for storage. Available studies suggest that the frozen storage process may lead to oxidative cleavage of the fat in human milk samples and lead to an increase in fishy flavor [22].

The 15 sensory profile descriptors of different human milk samples greatly differed in scores. To explore the possible reasons for the differences, sensory profiles of the human milk samples were grouped according to regions, total fat contents, and the stages of lactation period of donors, as shown in Figure 4A–C. In terms of geographical factors (Figure 4A), differences in sensory profiles were noticed in the human milk samples from different regions; however, considerable differences in sensory profiles were also seen among the human milk samples from the same region, especially Jiangsu and Anhui. Therefore, we preliminarily concluded that regional differences might not be the main reason for the differences in the sensory attribute strength among the human milk samples. Distinct sensory profile differences were also observed in the human milk samples with different fat contents (low, medium, and high), and surprisingly, the samples with a similar fat content exhibited analogous sensory profiles (Figure 4B). This observation indicated that fat content could be a crucial factor influencing the sensory characteristics of human milk, which was consistent with the findings in former research that fat content can affect the flavor of dairy products [31]. In terms of lactation period (Figure 4C), although there were some differences in the sensory profile of human milk samples from the same lactation period range, human milk samples from different lactation period ranges exhibited similar sensory profile patterns with that of human milk samples divided by fat content. That is, human milk with high lactation period and with a high fat content have similar sensory profile, and human milk with a low to medium lactation period and with low to medium fat content was the same. This may be related to the increase in fat content with the increase in the lactation period [39].

Therefore, using the established sensory profile descriptors and their quantitation analysis, the sensory characteristics of the human milk could be properly represented, and different human milk samples could also be effectively distinguished. Compared with the differences in region and lactation period, the fat content of human milk could be the more important reason for the difference in sensory attributes of human milk.

## 4. Conclusions

In summary, based on ISO 11035, the comprehensive sensory characteristics of human milk flavor were obtained by the sensory panel, and the milk flavor wheel was mapped for the first time. The M-value and multivariate statistical method were used to delete and merge the descriptors, and finally, a vocabulary consisting of 15 representative sensory profile descriptors was established to accurately describe the aroma, flavor, and mouthfeel of the human milk samples: fishy, dairy fat, metallic/iron, cooked, flour, dairy-sweet, and grassy/green for aroma; sweet, umami, creamy, fishy, and boiled milk for flavor; silky, fattiness, and mouthfulness for mouthfeel. The reference and the corresponding intensity were also set for each of the 15 sensory profile descriptors. Based on the established sensory vocabulary, both qualitative and quantitative sensory evaluation was conducted on the human milk samples, and the results indicated that the vocabulary could represent the sensory characteristics of different human milk samples. Further investigation exposed that the fat content in human milk might be a more important factor that may influence the sensory profiles of the human milk, compared with sample geographical regions and lactation periods.

To the best of our knowledge, this is the first study to report the construction of a flavor wheel for human milk. This study offers basic sensory data and general guidance for the study of human milk flavor, which may help people to comprehensively understand the human milk sensory composition and also provide clues for flavor-guide research and infant formula milk production that mimics the flavor of human milk. To be clear, our flavor wheel does not yet completely cover all the possible sensory characteristics in human milk. The study was limited by the number of human milk samples that could be evaluated by sensory evaluators in a single evaluation experiment, and meanwhile, many other factors that may influence the sensory properties of human milk could not be included, such as maternal or infantile illness, parity, diet, maternal exercise, etc. [40]. Therefore, just like the flavor wheel of the other foods, the human milk flavor wheel needs to be improved further in future research. Moreover, it is worth noting that all sensory results of this study were based on adult analysis of human milk samples, as it was not possible to use infants for sensory evaluation. Therefore, subsequent studies should clarify the differences in sensory perception of human milk between infants and adults, which could be a potential limitation of human milk and infant formula flavor studies.

## Figures and Tables

**Figure 1 nutrients-14-05387-f001:**
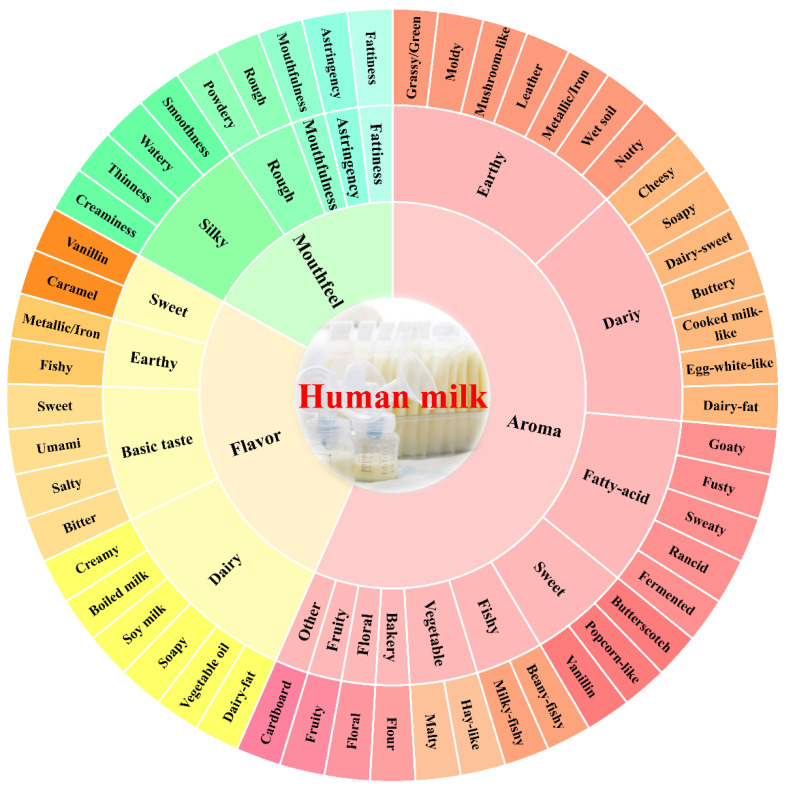
Human milk flavor wheel.

**Figure 2 nutrients-14-05387-f002:**
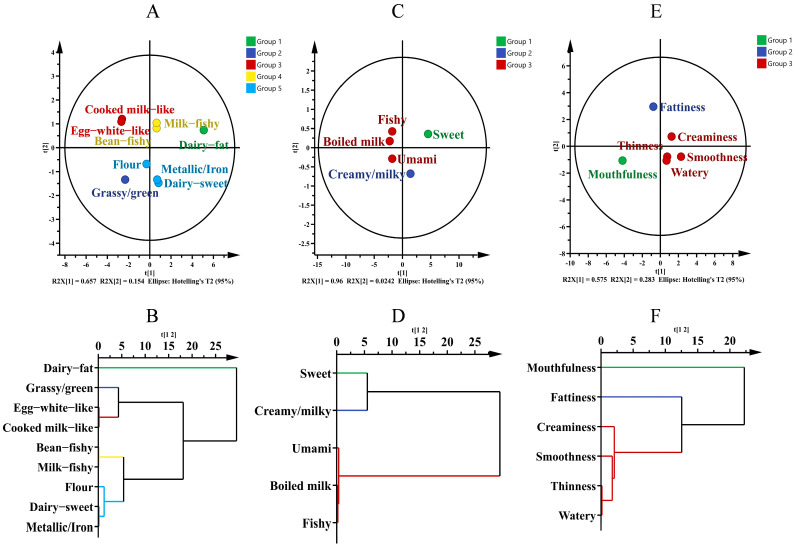
Principal component analysis (PCA) and hierarchical cluster analysis (HCA) of the sensory profile descriptors of human milk. (**A**,**B**) aroma, (**C**,**D**) flavor, (**E**,**F**) mouthfeel.

**Figure 3 nutrients-14-05387-f003:**
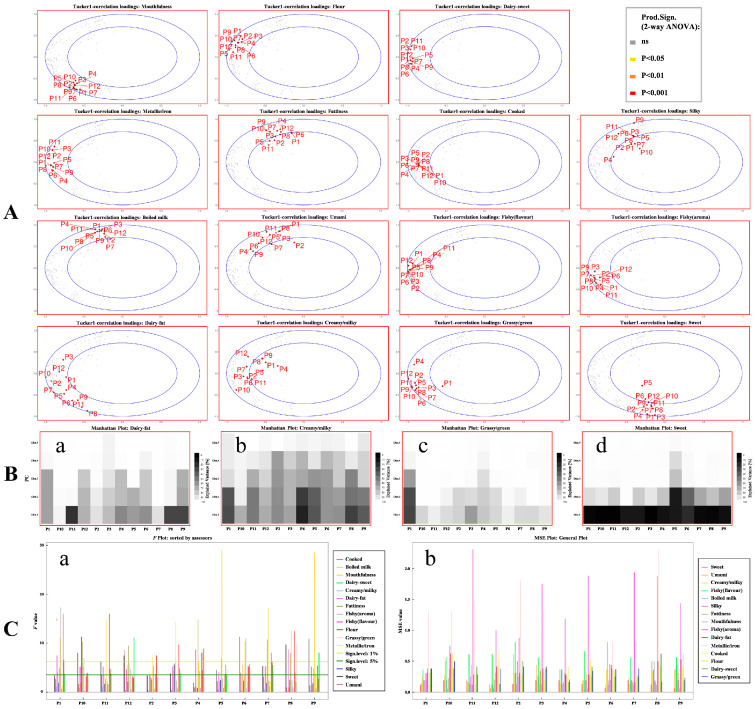
Evaluation of the sensory panel performance. (**A**,**B**) consistency, (**C**(**a**)) discrimination ability, (**C**(**b**)) repeatability. (p: panelists). (**B**(**a**–**d**)) Manhattan plot of dairy-fat, creamy, grassy/green and sweet, respectively).

**Figure 4 nutrients-14-05387-f004:**
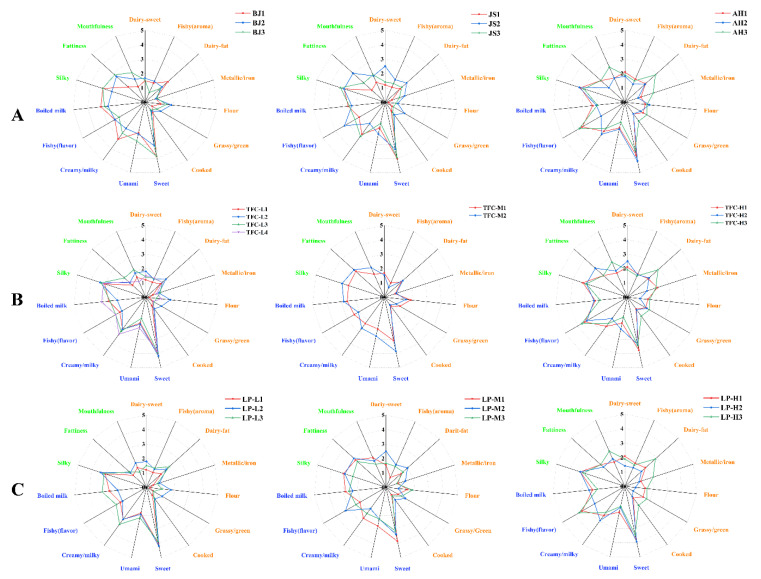
The sensory profile spider-web diagrams of human milk samples. (**A**) human milk samples from different geographical regions (BJ: Beijing; JS: Jiangsu; AH: Anhui), (**B**) human milk samples with different total fat contents (TFC), (**C**) human milk samples with different lactation periods (LP). (LP-H: high lactation period; LP-M: medium lactation period; LP-L: low lactation period; TFC-H: high total fat content, TFC-M: medium total fat content; TFC-L: low total fat content; BJ: Beijing; JS: Jiangsu; AH: Anhui). The sensory intensity of human milk was quantitatively scored on a scale of 0 to 5 with 0.5 increments, where “0” means that the flavor was not felted, and “5” means that the flavor was strong.)

**Table 1 nutrients-14-05387-t001:** The information of 18 human milk samples.

Samples ^a^	Regions	Lactation Periods (Month)	Content (g/100 mL)				Energy
			Fat	Protein	Carbohydrate	TS ^b^	(kcal/100 mL)
M1	Beijing	3	2.13 ± 0.09	0.77 ± 0.05	5.62 ± 0.19	8.83 ± 0.26	46.00 ± 1.63
M2	Beijing	4	3.07 ± 0.09	0.97 ± 0.05	6.20 ± 0.14	10.50 ± 0.22	58.00 ± 1.41
M3	Beijing	6	4.53 ± 0.05	0.77 ± 0.05	6.43 ± 0.05	11.90 ± 0.00	71.00 ± 0.00
M4	Beijing	2	1.80 ± 0.14	0.67 ± 0.05	4.27 ± 0.54	8.17 ± 0.50	40.67 ± 2.05
M5	Beijing	6	4.50 ± 0.08	1.17 ± 0.12	7.27 ± 0.05	13.13 ± 0.12	75.67 ± 0.47
M6	Beijing	3	2.33 ± 0.05	1.23 ± 0.05	4.57 ± 0.09	9.13 ± 0.12	50.45 ± 0.36
M7	Jiangsu	4	2.40 ± 0.00	0.93 ± 0.05	6.30 ± 0.08	9.73 ± 0.05	51.67 ± 0.94
M8	Jiangsu	2	2.37 ± 0.05	0.80 ± 0.08	5.63 ± 0.45	8.50 ± 0.75	50.00 ± 4.55
M9	Jiangsu	5	3.43 ± 0.12	0.80 ± 0.02	8.00 ± 0.08	12.43 ± 0.12	67.33 ± 1.25
M10	Jiangsu	4	4.57 ± 0.05	1.10 ± 0.03	4.40 ± 0.05	11.77 ± 0.05	68.33 ± 0.47
M11	Jiangsu	3	3.33 ± 0.05	0.53 ± 0.05	4.87 ± 0.05	10.13 ± 0.05	56.33 ± 0.47
M12	Jiangsu	4	3.40 ± 0.08	0.83 ± 0.05	4.70 ± 0.10	10.53 ± 0.09	57.67 ± 0.47
M13	Anhui	2	1.83 ± 0.12	0.63 ± 0.05	6.47 ± 0.05	9.10 ± 0.08	45.33 ± 1.25
M14	Anhui	5	5.30 ± 0.37	0.63 ± 0.05	5.60 ± 0.14	11.87 ± 0.42	75.00 ± 3.74
M15	Anhui	1	2.90 ± 0.28	0.97 ± 0.05	6.30 ± 0.08	10.40 ± 0.29	56.33 ± 2.62
M16	Anhui	5	4.33 ± 0.12	0.83 ± 0.12	4.63 ± 0.05	11.53 ± 0.05	66.33 ± 0.47
M17	Anhui	6	3.20 ± 0.24	1.00 ± 0.00	5.77 ± 0.05	10.17 ± 0.26	57.33 ± 2.49
M18	Anhui	4	3.00 ± 0.00	1.10 ± 0.05	4.65 ± 0.04	10.43 ± 0.08	58.00 ± 0.36

^a^ M: Human milk sample; ^b^ TS: total solids.

**Table 2 nutrients-14-05387-t002:** Geometric mean *M* and variance of sensory descriptors of human milk.

No.	Descriptors ^a^	*F* ^b^	*I* ^c^	*M* ^d^	Variance
Aroma					
1	**dairy fat**	0.9167	0.4741	0.6592	0.1931
2	**metallic/iron**	0.8796	0.4519	0.6304	0.1550
3	**flour**	0.8056	0.3426	0.5253	0.1391
4	**dairy-sweet**	0.7593	0.3278	0.4989	0.1250
5	**grassy/green**	0.7222	0.2926	0.4597	0.5048
6	**egg-white-like**	0.7315	0.2648	0.4401	0.1658
7	**cooked-milk-like**	0.6852	0.2667	0.4275	0.1219
8	**milky-fishy**	0.6204	0.2815	0.4179	0.2214
9	**beany-fishy**	0.5463	0.2537	0.3723	0.1969
10	**sweaty**	0.5741	0.2204	0.3557	0.0692
11	**rancid**	0.3704	0.1981	0.2709	0.0480
12	vanillin	0.3426	0.1370	0.2167	0.0552
13	soapy	0.3704	0.1222	0.2128	0.1127
14	goaty	0.3241	0.1370	0.2107	0.0691
15	wet soil	0.3056	0.1444	0.2101	0.0525
16	cheesy	0.2963	0.1167	0.1859	0.0664
17	buttery	0.2870	0.1204	0.1859	0.1185
18	butterscotch	0.2778	0.0944	0.1620	0.0247
19	hay-like	0.2870	0.0778	0.1494	0.0463
20	floral	0.2407	0.0778	0.1368	0.0509
21	cardboard	0.2407	0.0759	0.1352	0.0434
22	moldy	0.2500	0.0704	0.1326	0.0336
23	fusty	0.2222	0.0722	0.1267	0.0432
24	popcorn-like	0.1759	0.0870	0.1237	0.0552
25	fermented	0.1944	0.0778	0.1230	0.0478
26	mushroom-like	0.2130	0.0648	0.1175	0.0192
27	leather	0.1389	0.0852	0.1088	0.0130
28	nutty	0.2037	0.0519	0.1028	0.0377
29	fruity	0.1481	0.0519	0.0876	0.0501
30	malty	0.1667	0.0426	0.0843	0.0233
Flavor					
31	**sweet**	1.0000	0.7528	0.8676	0.1161
32	**creamy**	0.9028	0.4833	0.6606	0.0417
33	**umami**	0.6667	0.2056	0.3702	0.0583
34	**fishy**	0.5694	0.2056	0.3421	0.1173
35	**boiled milk**	0.4167	0.1944	0.2846	0.0590
36	vanillin	0.2500	0.1361	0.1845	0.0177
37	salty	0.3056	0.0972	0.1724	0.0397
38	vegetable oil	0.2361	0.1222	0.1699	0.1474
39	bitter	0.2361	0.1167	0.1660	0.0174
40	metallic/iron	0.1944	0.1083	0.1451	0.0347
41	caramel	0.2500	0.0778	0.1394	0.1752
42	soy milk	0.1667	0.0722	0.1097	0.1404
43	dairy fat	0.1806	0.0639	0.1074	0.1462
44	soapy	0.0833	0.0556	0.0680	0.1590
Mouthfeel					
45	**smoothness**	0.9722	0.4778	0.6815	0.3488
46	**fattiness**	0.9444	0.4194	0.6294	0.1277
47	**creaminess**	0.8472	0.4528	0.6194	0.0814
48	**watery**	0.8750	0.4167	0.6038	0.3472
49	**thinness**	0.7500	0.4319	0.5692	0.1728
50	**mouthfulness**	0.5972	0.2778	0.4522	0.0386
51	powdery	0.3750	0.1639	0.2479	0.0386
52	astringency	0.3056	0.1278	0.1976	0.3858
53	rough	0.1111	0.0333	0.0609	0.0104

^a^ Bolded descriptors means that the *M* value of these descriptors were higher than the mean value (aroma descriptors, 0.2623; flavor descriptors, 0.2705; and mouthfeel descriptors, 0.4513) ^b^
*F* represents the ratio of the times that a specific descriptor was actually mentioned to its total times that it was possibly mentioned. ^c^
*I* is the ratio of the sum of actual intensities for a descriptor given by the evaluation panel over its maximum possible intensity. ^d^
*M* is geometric mean, $M=\sqrt{F*I}$.

**Table 3 nutrients-14-05387-t003:** Definitions, references, and the corresponding intensities of the selected sensory profile descriptors for human milk.

Sensory Modalities	Descriptors	Definition	References	Intensity ^f^
aroma	fishy (aroma)	the aromatics associated with fresh fish	dried fillet ^a^	3
	dairy fat	the oily aromatics reminiscent of milk or dairy fat	whipping cream ^b^	3
	flour	the aromatics associated with wheat flour	wheat flour (pure)	3
	metallic/iron	the aromatics associated with iron rust	iron rust (IR) ^a^	0.1 g IR/1 g water = 1.5
				0.2 g IR/1 g water = 3
				0.4 g IR/1 g water = 4
	cooked	the combination of brown flavor notes and aromatics associated with heated milk	milk heated to 85 °C for 45 min ^c^	50% heated milk = 1.5
				100% heated milk = 3
	dairy-sweet	the sweet aromatics associated with fresh dairy products	vitamin D milk ^b^	50% vitamin D milk = 1.5
				pure vitamin D milk = 3
	grassy/green	the aromatics associated with lawns after it’s been mowed	hexanal	10 μL/100 mL = 1
				30 μL/100 mL = 3
				40 μL/100 mL = 4
flavor	sweet	fundamental taste sensation of which sucrose is typical	sucrose ^d^	1.06% sucrose = 1.5
				2.12% sucrose = 3
	umami	fundamental taste sensation of which monosodium glutamate (MSG) is typical	MSG ^d^	0.05% MSG = 1
				0.15% MSG = 3
				0.20% MSG = 4
	creamy	the flavor reminiscent of milk or dairy fat	milk fat (MF) ^d^	50% MF = 1.5
				pure MF = 3
	fishy (flavor)	the flavor associated with juice of canned fish	canned tuna juice (CTJ) ^c^	20% CTJ = 2
				30% CTJ = 3
				40% CTJ = 4
	boiled milk	the flavor reminiscent of heated milk	milk heated to 85 °C for 45 min ^c^	50% heated milk = 1.5
				100% heated milk = 3
mouthfeel	silky	degree of smoothness felt in the mouth	pure milk ^e^	3
	fattiness	the perceived of creaminess-like in the mouth	pure milk ^e^	3
	mouthfulness	the feel of coating, long-lasting, thickness	pure milk ^e^	3

^a^ Objects were used as references in reference to the study by Yang et al. ^b^ Objects were used as references in reference to the study by Heisserer et al. ^c^ Objects were used as references in reference to the study by Drake et al. ^d^ Objects were used as references in reference to the study by Mastorakou et al. ^e^ Since no suitable references to these mouthfeel attributes was found for existing studies, and pure milk has a similar mouthfeel, therefore, pure milk was used as a reference for sensory training. ^f^ Without special instructions, all references were diluted or dissolved by ultra-pure water.

## Data Availability

Not applicable.

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
