# Peer review of "Flavor Wheel Construction and Sensory Profile Description of Human Milk"

_nutrients, 2022, doi:10.3390/nu14245387_

Round 1
Reviewer 1 Report
This paper describes a study designed to establish a sensory profile for human milk. While the logic for doing this in terms of optimisation of quality of infant formula is apparent, in general there is not a very strong case made for the rationale for the study, which should be made more explicit (it is only mentioned briefly in the introduction and in the conclusions).
A major assumption that appears to underpin the study is that infants and adults have the same sensory perception of the milk. While it is obviously difficult to prove this one way or the other, the authors should make some effort to consider this, provide any evidence that might exist in the literature, and acknowledge it as a potential limitation of their work.
Some further minor points for consideration are as follows:
· Is 100-150 mL of milk typical for an expression? Were there only one set of samples (i.e., one collection)? It seems this volume is quite low to cover all the testing and tasting described.
· The use of the 18 samples is occasionally confusing. For example, on line 144 are the six randomly sampled milks a subset of the 18 then evaluated as described on line 158? Also, on line 209 were there three samples from each region or three samples randomly selected from across the regions? Are these numbers of samples accepted as sufficient in power to draw the conclusions reached?
· In Table 2, it needs to be explained why some parts of the table are in bold
· On line 190, the use of a 50ml glass vial is not clear – how is that of use for samples as possibly diverse in size and shape as dried filled and MSG?
· On line 295, a better word or description for ‘mouthfulness’ is needed
· In some of the interpretation of sensory results, more discussion relative to human milk components is needed. For example, lactose is not mentioned in relation to sweetness, despite being the most dominant component of human milk
· The caveats introduced on lines 405-410 about handling may be overstated – is there evidence that oxidation would be such an issue? Would some direct comparison of fresh and frozen human milk not have been useful to conduct?
· On line 457-459, the potential importance of maternal diet is not considered enough
· Line 464: what is meant by flavour-guide research?
Author Response
A list of corrections is attached, please have a check.

Reviewer 2 Report
The manuscript nutrients-2074708 ‘Flavor wheel construction and sensory profile description of human milk’, Nutrients
The manuscript contains interesting research results, and I congratulate you on the implementation of this study. However, in my opinion, the manuscript requires additional revisions that would allow an even more comprehensive approach to the topic described. Below are my proposals, in order to refine the manuscript before its publication:
1. Line 125-140
Whether they were trained experts?
2. Line 141 and 180
Exactly how many samples have been tested by experts. This should be specified
3. Line 141 and 180
In how many repetitions the survey was conducted? This should be specified
4. Line 141 and 180
Could you explant the choice of type of scale in both tests? Why you do not use 0-10 point scale for attributes intensity evaluation?
5. Line 167
Why you did not evaluated appearance of samples?
6. Line 196
Please verify the definition of umami, because it is also basic taste sensation as sweet one, described above.
7. Line 491
Please prepare the list of references according to the journal requirements
Author Response

(The authors gave the same response as above.)
